# Genetic and Transcriptomic Characteristics of RhlR-Dependent Quorum Sensing in Cystic Fibrosis Isolates of *Pseudomonas aeruginosa*

Kyle L. Asfahl,[a] Nicole E. Smalley,[a] Alexandria P. Chang,[b] Ajai A. Dandekar[a,b]

[a]Division of Pulmonary, Critical Care, and Sleep Medicine, Department of Medicine, University of Washington, Seattle, Washington, USA
[b]Department of Microbiology, University of Washington, Seattle, Washington, USA

**ABSTRACT** In people with the genetic disease cystic fibrosis (CF), bacterial infections involving the opportunistic pathogen *Pseudomonas aeruginosa* are a significant cause of morbidity and mortality. *P. aeruginosa* uses a cell-cell signaling mechanism called quorum sensing (QS) to regulate many virulence functions. One type of QS consists of acyl-homoserine lactone (AHL) signals produced by LuxI-type signal synthases, which bind a cognate LuxR-type transcription factor. In laboratory strains and conditions, *P. aeruginosa* employs two AHL synthase/receptor pairs arranged in a hierarchy, with the LasI/R system controlling the RhlI/R system and many downstream virulence factors. However, *P. aeruginosa* isolates with inactivating mutations in *lasR* are frequently isolated from chronic CF infections. We and others have shown that these isolates frequently use RhlR as the primary QS regulator. RhlR is rarely mutated in CF and environmental settings. We were interested in determining whether there were reproducible genetic characteristics of these isolates and whether there was a central group of genes regulated by RhlR in all isolates. We examined five isolates and found signatures of adaptation common to CF isolates. We did not identify a common genetic mechanism to explain the switch from Las- to Rhl-dominated QS. We describe a core RhlR regulon encompassing 20 genes encoding 7 products. These results suggest a key group of QS-regulated factors important for pathogenesis of chronic infections and position RhlR as a target for anti-QS therapeutics. Our work underscores the need to sample a diversity of isolates to understand QS beyond what has been described in laboratory strains.

**IMPORTANCE** The bacterial pathogen *Pseudomonas aeruginosa* can cause chronic infections that are resistant to treatment in immunocompromised individuals. Over the course of these infections, the original infecting organism adapts to the host environment. *P. aeruginosa* uses a cell-cell signaling mechanism termed quorum sensing (QS) to regulate virulence factors and cooperative behaviors. The key QS regulator in laboratory strains, LasR, is frequently mutated in infection-adapted isolates, leaving another transcription factor, RhlR, in control of QS gene regulation. Such isolates provide an opportunity to understand Rhl-QS regulation without the confounding effects of LasR, as well as the scope of QS in the context of within-host evolution. We show that a core group of virulence genes is regulated by RhlR in a variety of infection-adapted LasR-null isolates. Our results reveal commonalities in infection-adapted QS gene regulation and key QS factors that may serve as therapeutic targets in the future.

**KEYWORDS** adaptation, cystic fibrosis, gene regulation, quorum sensing, transcriptomics

Address correspondence to Ajai A. Dandekar, dandekar@uw.edu.

The authors declare no conflict of interest.

Persistent infection with the environmental bacterium and opportunistic pathogen *Pseudomonas aeruginosa* is a common clinical problem. Burn wounds, indwelling catheters, and the airways of people with the genetic disease cystic fibrosis (CF) all

provide ecological niches suited to *P. aeruginosa* infection (1, 2). Acute infections begin with motile *P. aeruginosa* expressing a variety of secreted toxins, proteases, and virulence factors to establish infection, aided by intrinsic resistance to several classes of antibiotics (3). A transition follows, where *P. aeruginosa* adapts to favor a biofilm lifestyle that protects the pathogen from host immune factors (3–5). Increased host inflammation and fluid viscosity further aid biofilm construction, particularly in the CF lung, where *P. aeruginosa* infection is associated with morbidity and mortality (6, 7). Ultimately, *P. aeruginosa* strains adapted to chronic infections persist despite aggressive treatment.

A substantial fraction of the virulence factor program of *P. aeruginosa* is regulated via a cell-cell communication system termed quorum sensing (QS), where a diffusible signal binds a cognate receptor that is a transcriptional activator, thereby allowing monitoring of population density and a coordinated transcriptional response (8, 9). QS provides a mechanism for *P. aeruginosa* to specifically modulate virulence gene expression based on environmental constraints within the host (8, 10). *P. aeruginosa* has two complete QS systems that use acyl-homoserine lactone (AHL) signals: LasI synthesizes $N$-3-oxo-dodecanoyl-homoserine lactone ($3OC_{12}$-HSL), which is bound by LasR, and RhlI synthesizes $N$-butanoyl homoserine lactone ($C_4$-HSL), which is bound by RhlR (Fig. 1) (11). An orphan AHL receptor, QscR, binds to $3OC_{12}$-HSL and negatively regulates LasIR through activation of a single linked operon (PA1897-91) (12–14). Finally, a non-AHL QS circuit in *P. aeruginosa* uses alkyl-quinolone (AQ) signals that are bound by their receptor, the LysR-type transcription factor PqsR (also called MvfR) and is termed the *Pseudomonas* quinolone signal (PQS) system (15). Most of the *P. aeruginosa* QS circuity was defined in strain PAO1 and has been described as a hierarchy with LasR at the top (11). LasR activates *lasI*, *lasR*, *rhlI*, *rhlR*, genes for enzymes involved in AQ biosynthesis (*pqsABCDE* and *pqsH*), and the PQS receptor gene *pqs* (*mvfR*) (11). There are additional layers of regulation: RhlR directly represses transcription of *pqsABCDE*, and PqsE enhances RhlR activity through a protein-protein interaction (16–19). Together, these *P. aeruginosa* QS systems allow concerted population-wide regulation of more than 200 genes in response to cell density (20, 21).

A complex interplay of selective pressures contributes to diversification of *P. aeruginosa* during the shift from acute to chronic infection, and the QS systems of *P. aeruginosa* appear to be common targets of selection. The gene encoding LasR is often mutated in isolates from chronic CF infections, yet mutations in RhlR are rare (22–25). This observation suggested that RhlR may function independently of LasR in some isolates, in contrast to the LasR-dominated circuitry exhibited in laboratory strains (Fig. 1). This notion is supported through observations of isolates with variant LasR alleles from the lungs of CF patients, where Rhl-QS appears to regulate production of elastase, rhamnolipid surfactants, and the toxic secondary metabolite pyocyanin (24).

We are interested in understanding the evolution of *P. aeruginosa* QS as strains become adapted to the CF lung and switch from LasR- to RhlR-dominated QS. The identification of such isolates also presents an opportunity: the hierarchical arrangement of QS in laboratory strains, where LasR is required for activation of Rhl-QS under most circumstances, has presented a roadblock to understanding the RhlR regulon. The scope of Rhl-QS-mediated gene regulation has been approached previously via transcriptome analysis of laboratory strains with mutations in *rhlR* or *rhlI*, through supplementation of synthetic $C_4$-HSL signals, or both (26, 27). These studies identified lists of genes that are generally considered to be subsets of QS targets regulated by LasR, and because LasR activation was instrumental in these designs, understanding the explicit importance of RhlR in their regulation remained elusive. We hypothesized that a core regulon for RhlR exists among isolates with a common mode of infection, that of the CF lung, and that a common genomic adaptation might explain the switch to Las-independent Rhl-dominated QS.

Here, we used a collection of CF isolates with naturally occurring, inactivating mutations in LasR to determine a core regulon for the QS receptor RhlR among 5 infection

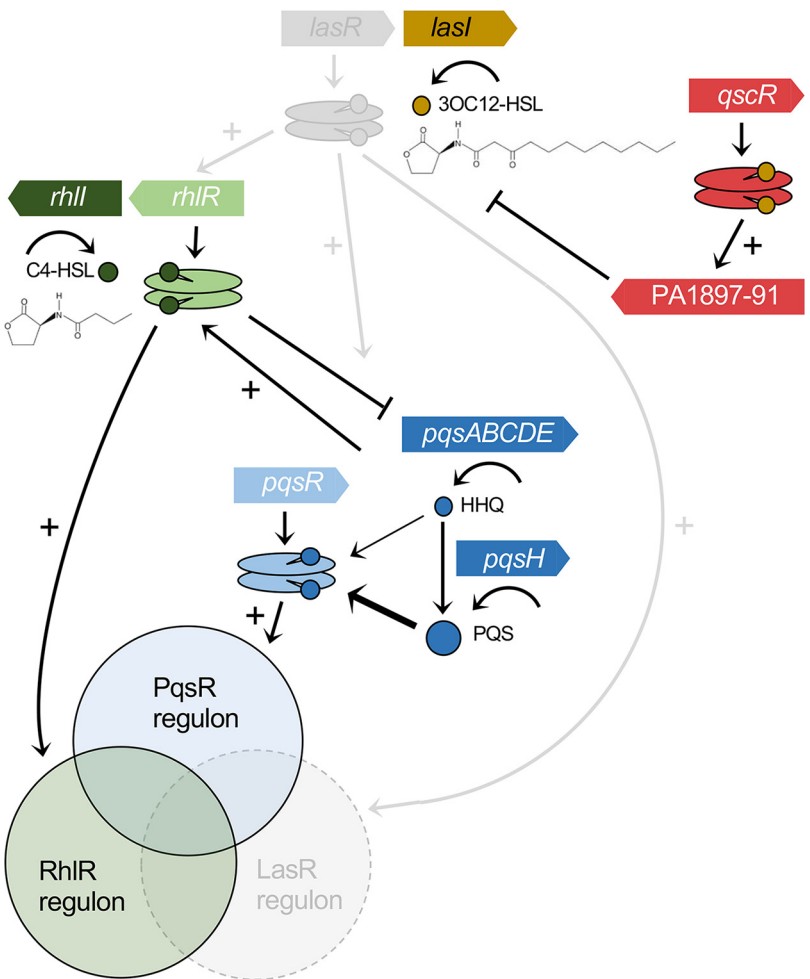

**FIG 1** Las-independent Rhl-QS circuitry in *P. aeruginosa*. In laboratory strains, LasR activation is required for downstream activation of the Rhl-QS (green) and PQS (blue) circuitry. The CF isolates analyzed in this study harbor deleterious mutations that inactivate LasR (gray components) yet still use RhlR and the PQS system to regulate overlapping sets of target genes. The orphan regulator QscR (red) binds the LasI-generated signal and activates a single linked operon.

isolates. We show that these isolates do not respond to the $3OC_{12}$-HSL signal through LasR but are still able to use the Rhl-QS system to activate RhlR target promoters in a density-dependent manner, control pyocyanin production, and augment extracellular proteolysis. We conducted a comparative genomic analysis to find an identifiable switch that could yield Las-independent Rhl-QS. We used deletion of *rhlR* and supplementation of synthetic QS signals to yield 5 parallel RhlR-off and RhlR-on conditions for use in a transcriptome sequencing (RNA-seq)-based RhlR transcriptome analysis for each isolate. We found that among 227 genes regulated by RhlR in at least 1 isolate in our cohort, just 20 genes make up the core RhlR regulon.

## RESULTS

**Isolate selection.** We identified CF isolates with naturally occurring *lasR* variant alleles through a screen of a collection of >2,500 *P. aeruginosa* strains isolated from the lungs of children enrolled in the Early *Pseudomonas* Infection Control (EPIC) observational study (28). Our group has previously reported on the prevalence of nonsynonymous *lasR* variants in this collection (~22%) with an initial characterization of the QS activity in 31 isolates harboring unique *lasR* variant alleles (24).

For the present study, we sought to understand Las-independent Rhl-QS, so our initial screen criteria required the isolates to have loss-of-function *lasR* alleles while also

**TABLE 1** CF isolate characteristics

| | AHL production ($\mu$M)[a] | | LasR mutation | |
|---|---|---|---|---|
| Isolate | C$_4$-HSL | 3OC$_{12}$-HSL | Nucleotide | Amino acid |
| E104 | 17.73 | 0.39 | A532G | T178A |
| E113 | 3.45 | 0.05 | T55C | W19R |
| E125 | 13.43 | 0.07 | C280T | Q94Stop |
| E131 | 26.54 | 0.14 | A580G | S194G |
| E167 | 7.44 | 0.01 | G339del | Frameshift |

[a]Planktonic AHL determinations at 18 h from reference 24. PAO1 produced approximately 9 $\mu$M C$_4$-HSL and PAO1 *lasR* produced <2 $\mu$M in that study. AHL, acyl-homoserine lactone.

producing C$_4$-HSL, as reported in our previous study (24). Of the isolates we were interested in, we required that the isolates accept transcriptional fusion reporter plasmids and be amenable to genetic manipulation. We selected 5 such isolates, all of which were from different patients (Table 1). These isolates harbor a range of inactivating mutations that include premature stop codons (E125), frameshifts (E167), or residue changes that interrupt the DNA-binding (E104 and E131) or signal-binding (E113) domains of LasR (24).

**RhlR controls gene promoters and phenotypes associated with virulence in a cohort of CF isolates.** First, we queried whether QS transcriptional activity was in fact independent of LasR using transcriptional fusion reporter plasmids. We asked if the *lasR* alleles in our CF isolates retained LasR activity. To do so, we measured green fluorescent protein (GFP) fluorescence from cells transformed with a P$_{lasI}$-*gfp* reporter. The *lasI* promoter is LasR-activated, and, consistent with the idea that the mutations in *lasR* were inactivating, we found that fluorescence was negligible in each of our clinical isolates compared to PAO1. Because a lack of gene activation could reflect inadequate signal concentrations rather than nonfunctional protein, we also asked if addition of the LasR signal 3OC$_{12}$-HSL increased fluorescence in any of the isolates (Fig. 2A). It did not.

Next, to measure transcription from a RhlR-regulated promoter, we measured GFP fluorescence using a P$_{rhlA}$-*gfp* reporter. The promoter for *rhlA* is strictly controlled by RhlR (29, 30). Despite mutations in *lasR* in our CF isolates, each strain showed activation of the *rhlA* promoter (P$_{rhlA}$-*gfp*) that was significantly greater than that in PAO1 containing a *lasR* deletion (Fig. 2B). We then queried whether the timing of induction of the *rhlA* promoter was advanced compared to that in PAO1 by following activation over time; no strain was significantly advanced in P$_{rhlA}$-*gfp* induction (Fig. S1). Together, these results indicated that in these CF isolates, LasR is inactive and unresponsive to its signal, but RhlR remains active as a QS transcriptional regulator.

In an effort to understand the range of QS-controlled phenotypes in our CF isolate cohort, we examined production of the redox-active pigment and secreted toxin pyocyanin, and extracellular proteolysis. First, we determined the level of pyocyanin produced by our isolates and their isogenic *rhlR* mutants. Production of pyocyanin was elevated in every isolate compared to the laboratory strain PAO1, ranging from roughly 2-fold that of PAO1 in strain E167 to 25-fold in strain E131 (Fig. 2C). Regardless of the level produced by each wild-type isolate, deletion of *rhlR* in each isolate background reduced pyocyanin to levels lower than that of wild-type PAO1, and similar to that of the reference QS null mutant PAO1 *lasR rhlR*. We then expanded our analysis to extracellular proteolysis, using skim-milk agar plates (31). In the laboratory strain PAO1, secreted proteases such as the QS-controlled factor LasB elastase diffuse away from the colony, producing a zone of reduced opacity with a radius that varies with regulation of protease. *lasR* and *lasR rhlR* mutants of PAO1 show a marked reduction in this radius of clearing while the *rhlR* mutant does not (Fig. 2D), reflecting the QS hierarchy of PAO1. We found that while the radius of clearing varied between individual wild-type isolates and relative to PAO1, deletion of *rhlR* significantly reduced this radius in each isolate (Fig. 2D). Together, these phenotypes indicated that despite harboring inactivating mutations in *lasR*, these isolates still use RhlR to regulate production of virulence factors.

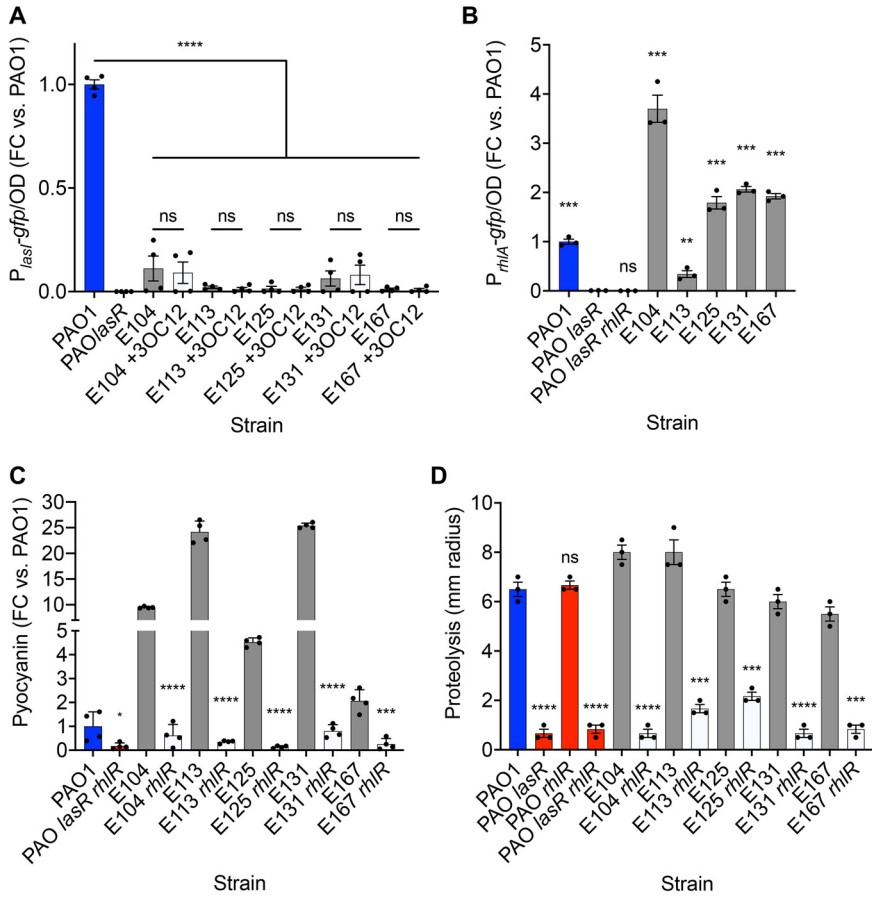

**FIG 2** Las-independent QS transcription and phenotypes. (A) pP$_{lasI}$-*gfp* reporter activity, normalized to OD$_{600}$ and scaled to PAO1. 3OC$_{12}$-HSL was added to a final concentration of 2 $\mu$M. (B) pP$_{rhlA}$-*gfp* reporter activity, normalized to OD$_{600}$ and scaled to PAO1. (C) Pyocyanin production after 18 h, normalized to OD$_{600}$ and scaled to PAO1. (D) Extracellular proteolysis, determined as the zone of clearing on skim milk agar after 48 h. Statistical tests were performed to discern differences between conditions as indicated (A), between PAO *lasR* and each strain (B), or between individual mutants and their isogenic wild-type parents (C and D). In all panels, the means, standard errors, data points, and results of unpaired *t* tests are shown ($n \geq 3$). ns, not significant; *, $P < 0.05$; **, $P < 0.01$; ***, $P < 0.001$; ****, $P < 0.0001$.

**Genomic analysis.** We sequenced and assembled complete, closed genomes of each isolate for comparative genomic analysis and for use as strain-specific transcriptomic mapping references. Genome sizes were all greater than that of PAO1 (6.26 Mbp), ranging from 6.67 Mbp for strain E131 to 6.89 Mbp for strain E125 (Table S1). The number of annotated features was also greater in each isolate than PAO1, with an average of 658 additional annotated features (Table S1). Strain E167 also harbored a stably maintained plasmid of approximately 47 kbp.

In an effort to understand the relatedness of our CF isolate cohort to other sequenced isolates of *P. aeruginosa*, we compared our isolate genomes to a curated collection of genomes on the publicly available IMG/MER database (32). We constructed a phylogeny of our isolates (5), E90 from our previous study (33), complete genomes associated with cystic fibrosis (10) or airways (5), and strains PAO1 and PA14 (21 strains total) (Fig. 3). We found our 5 isolates were distributed throughout the phylogeny. Isolates E113 and E131 were closely clustered with the epidemic CF infection strains described from Liverpool (LESB58) and Manchester (C3719), with our previously described strain E90, and laboratory strains PAO1 and PA14. On the other hand, isolates E104, E125, and E167 were more distantly related and separated from the phylogeny, similar to the Danish transmissible strain DK2 and the composite assembly strain PACS2, isolated from a 6-month-old with

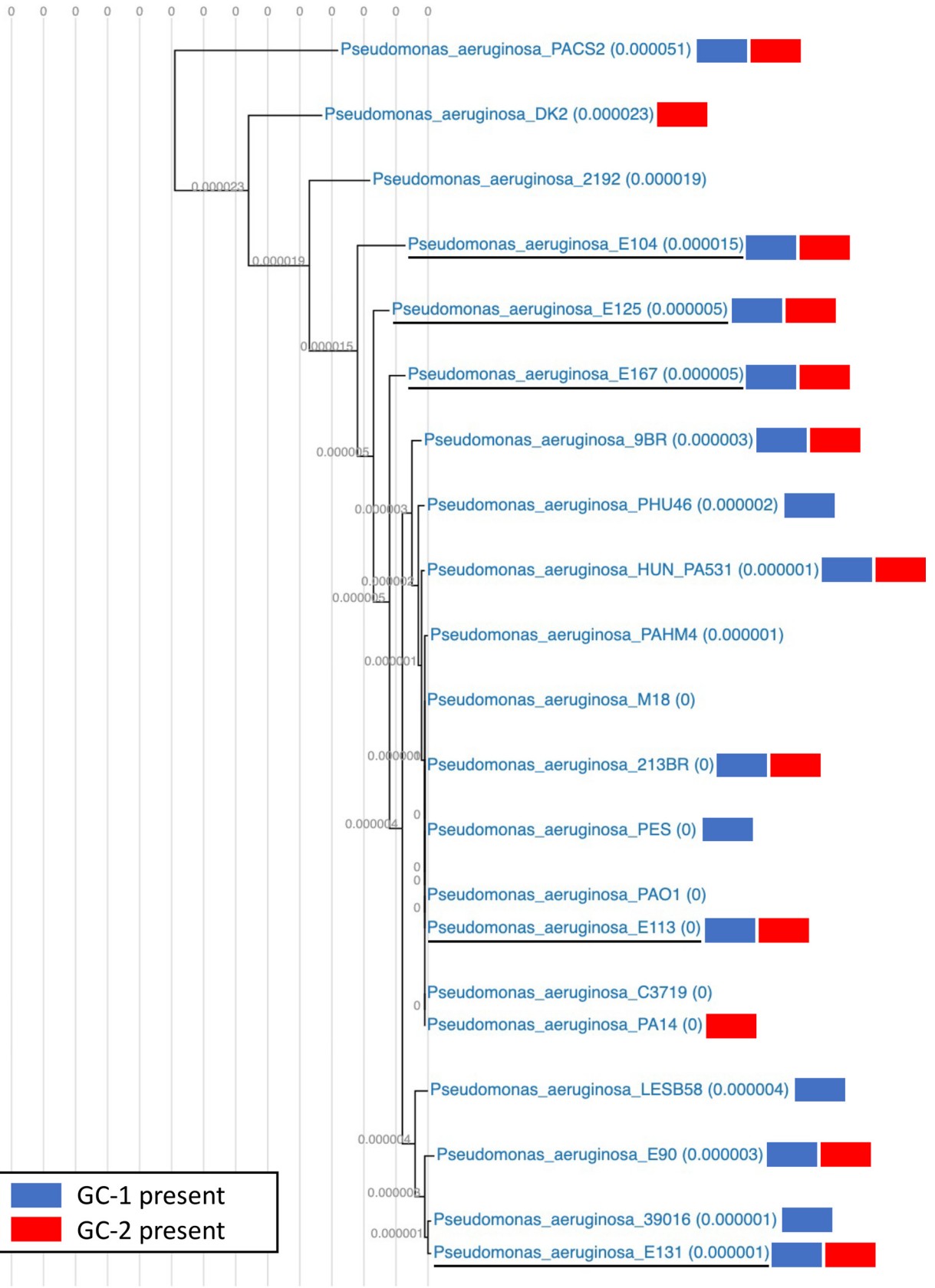

**FIG 3** Phylogenetic distribution of CF isolate cohort. CF isolates in this study (underlined) are well distributed in a phylogeny that includes laboratory strains and previously published genomes of *P. aeruginosa* isolates with airway or CF infection origins. Blue and red denote the presence of gene cassettes 1 and 2 (GC-1 and GC-2), respectively, in the genomes of individual isolates.

CF. Together, our phylogenetic analysis indicated that our 5-isolate cohort was genomically diverse and represented a variety of CF isolate origins.

Previous investigations have indicated the genome of *P. aeruginosa* may experience purifying selection during adaptation to the CF lung, with selection against certain acute virulence factors (22, 34). To gain an understanding of how the contents of our cohort isolate genomes compare to both PAO1 and previously published genomic analyses of CF isolates, we conducted a pangenomic analysis. We found a total of 7,156 genes in the pangenome of our isolate cohort and PAO1, with a core genome of 5,353 genes present in all six strains. We found deletions or inactivating mutations in 197 genes common among the isolate cohort but normally present in laboratory strains (Table S2). Seventy-four of these genes were missing entirely from all 5 genomes. This group of commonly mutated or missing genes includes 16 genes annotated as transcriptional regulators, including *lasR*. We did not find mutations in other genes coding for QS receptors or signal synthesis (*lasI*, *rhlRI*, *qscR*, *pqsABCDE*, *pqsH*, and *pqsR*) or a trio of genes encoding products that directly regulate QS (*rsaL*, *qteE*, and *qslA*) (11, 35, 36). Mutation of other transcriptional regulators such as *mexT* may provide a pathway to active Rhl-QS in the absence of a functional LasR, a feature observed during *in vitro* evolution of PAO1 (37, 38), but we found no evidence of mutations in *mexT* in our study.

Other regulators found to be missing or mutated included those involved in amino acid catabolism (*amaR* and *dguR*), terpene utilization (*atuR*), and multidrug efflux and antibiotic susceptibility (*armR*, *oprR*, and *cmrA*). The list of commonly lost or mutated genes also included other genes known to be targeted by selection during long-term evolution in the CF lung (22, 25, 39): O-antigen biosynthesis (*hisF2H2*, *wzx*, and *wzy*), type III secretion system (T3SS) (*exsD*), exotoxin A (*xqhB*), and twitching motility (*pilA*). Genes involved in interspecific interactions with neighboring microbes were also mutated in all 5 isolates: T2SS (*xqhB*), T4SS (*pilB*, -*S*, and -*Y1* and *fimT*), T6SS (*tse6* and *tssC1/L1*), and the S-pyocin killing mechanism (*pyoS5*, *pys2*, and PA3866).

In addition to genes commonly lost or mutated relative to PAO1, we found 112 genes in common among our cohort that are not present in PAO1 (Table S3). Of these common non-PAO1 genes, 87% (98) are organized into cassettes that harbor annotations for integrative transfer (e.g., *pil* genes), along with some strain-specific gene content. These consist of two organized and conserved cassettes of 28 and 70 genes (gene cassettes 1 and 2 [GC-1 and GC-2]). Both cassettes are widespread within the strains of our analyzed phylogeny (Fig. 3). GC-1 includes genes coding for at least 3 putative regulators, mitomycin antibiotic biosynthesis proteins, a collection of acyl coenzyme A (acyl-CoA) modifying enzymes, and several hypothetical proteins. GC-2 harbors at least 15 Incl1-group conjugative plasmid-associated genes (PFGI-1-like, *parB*, and *pilL*, -*N*, -*P*, -*Q*, -*S*, -*U*, and -*M* are present), genes coding for 2 putative regulators, the mercuric resistance protein MerT, genes involved in T4SS (*virB4*), a T3SS effector Hop protein, and several hypothetical proteins.

**A core regulon for RhlR.** The primary goal of the present study was to discern commonalities among Rhl-QS gene regulation in CF-adapted *P. aeruginosa*. To this end, we evaluated the transcriptomes of each isolate at the entry to stationary phase (optical density at 600 nm [OD$_{600}$], 2.0) (Fig. 4A), using differential expression (DE) analysis between wild-type CF isolates and their isogenic *rhlR* deletion mutants. This is a point in growth that previous studies have shown to provide a reasonable census of QS activation among various strains (26, 40). We also added 3OC$_{12}$-HSL and C$_4$-HSL to the wild-type condition to avoid differences in gene regulation due to variable signal production (Table 1).

RhlR regulons varied in size from 28 (E125) to 185 genes (E113) (Table S4). A 5-way comparison revealed that just 20 genes were shared among all 5 RhlR regulons, representing a core regulon for RhlR among our cohort of CF isolates (Fig. 4B). This core RhlR regulon is detailed in Table 2. Genes in the core regulon include those coding for several known virulence factors that have been previously reported to be regulated by RhlR (26, 33). These include genes coding for rhamnolipid (*rhlA*, PA3479), elastase (*lasB*,

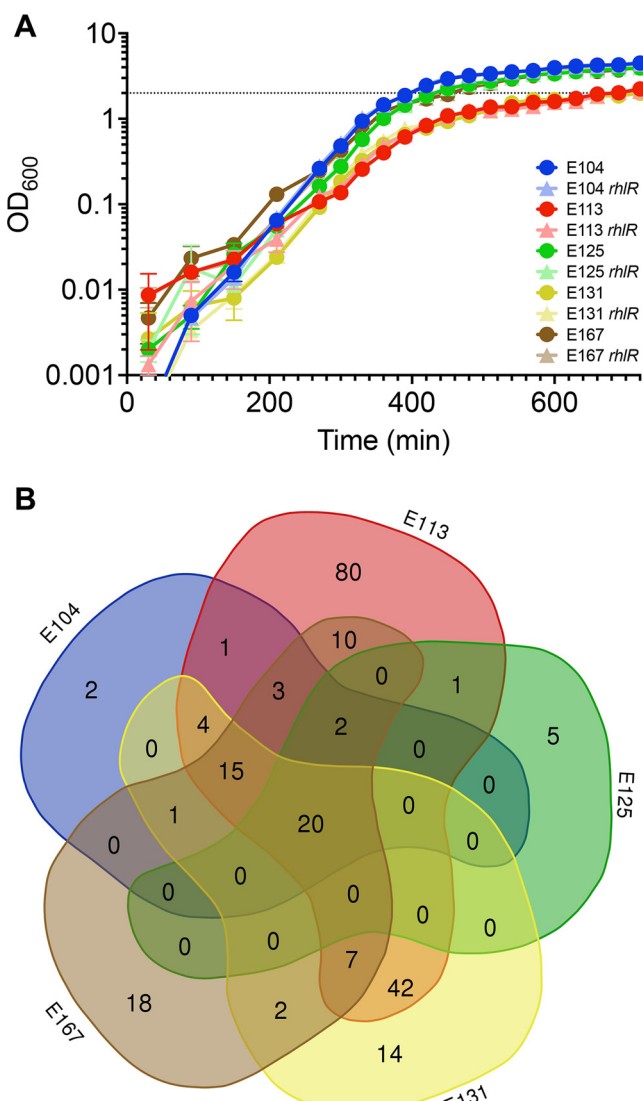

**FIG 4** RhlR activated genes: core analysis. (A) Growth of CF isolate strains is similar between isogenic wild-type and *rhlR* mutant pairs. Strains were grown from a starting $OD_{600}$ of 0.001 in LB+MOPS medium. Timing of total RNA harvest is shown with a dotted line ($OD_{600}$, 2.0). (B) Venn diagram displaying overlap of RhlR-activated genes in individual CF isolate regulons. Lobes are not scaled to size.

PA3724), hydrogen cyanide (*hcnABC*, PA2193 to -2195), and phenazine biosynthesis (*phzA1* and -*B1*, PA4210 and -4211). *rhlI* (PA3476) was found to be under strong positive regulation by RhlR, indicating that the autoinduction loop present in LasR/LasI in the Las system of other strains is also present in the Rhl system of our cohort isolates. The only gene in the core regulon annotated as a downstream transcriptional regulator is *vqsR* (PA2591). Making up nearly half of the core regulon, the RhlR-regulated genes spanning PA3326 to -3334 encode a recently described azetidomonamide (*aze*) biosynthesis regulon (PA3331 is RhlR regulated in 4/5 isolates) (41, 42). Finally, the core regulon also includes genes coding for two hypothetical proteins (PA1221 and PA4141). Expanding the core RhlR regulon list to include genes regulated by at least 4 of the 5 isolates (17 genes) adds the second phenazine locus (*phzA2B2C2*), LasA protease (*lasA*), and genes involved in the T6SS (*hsiA2*) and rhamnolipid biosynthesis (*rhlB*). The average fold change of the 20 core genes activated by RhlR was roughly 59-fold across the isolate cohort but varied considerably by isolate, ranging from 4.93 in E125 to 219.28 in E113 (Table 2).

**TABLE 2** Core RhlR regulated genes among CF isolates

| Locus tag[a] | Gene name[b] | Function[b] | CF isolate RhlR-induced fold change | | | | |
|---|---|---|---|---|---|---|---|
| | | | E104 | E113 | E125 | E131 | E167 |
| PA1221 | | Hypothetical protein | 5.19 | 20.07 | 2.71 | 4.03 | 5.78 |
| PA1869 | | Probable acyl carrier protein | 28.72 | 79.34 | 5.78 | 14.77 | 10.05 |
| PA2193 | *hcnA* | Hydrogen cyanide synthase HcnA | 23.85 | 65.55 | 9.66 | 8.91 | 39.16 |
| PA2194 | *hcnB* | Hydrogen cyanide synthase HcnB | 8.19 | 8.61 | 2.35 | 2.92 | 6.93 |
| PA2195 | *hcnC* | Hydrogen cyanide synthase HcnC | 3.43 | 8.52 | 2.06 | 2.74 | 5.53 |
| PA2591 | *vqsR* | Virulence and quorum sensing regulator VqsR | 13.63 | 10.87 | 6.89 | 3.34 | 14.16 |
| PA3326 | *azeA/clpP2* | AzeA/ClpP2 | 16.10 | 19.85 | 2.18 | 2.85 | 13.51 |
| PA3327 | *azeB* | Probable nonribosomal peptide synthetase | 21.13 | 102.17 | 2.83 | 5.12 | 36.02 |
| PA3328 | *azeC* | Probable FAD-dependent monooxygenase | 67.86 | 271.93 | 5.28 | 12.05 | 152.08 |
| PA3329 | *azeD* | Hypothetical protein | 18.71 | 83.98 | 3.03 | 8.22 | 45.88 |
| PA3330 | *azeE* | Probable short-chain dehydrogenase | 30.19 | 148.95 | 4.06 | 8.44 | 73.99 |
| PA3332 | *azeG* | Conserved hypothetical protein | 38.93 | 98.62 | 3.96 | 18.39 | 70.46 |
| PA3333 | *azeH/fabH2* | AzeH 3-oxoacyl-[acyl-carrier-protein] synthase III | 22.65 | 73.33 | 2.15 | 14.31 | 40.57 |
| PA3334 | | Probable acyl carrier protein | 20.35 | 103.48 | 3.10 | 17.32 | 36.68 |
| PA3476 | *rhlI* | Autoinducer synthesis protein RhlI | 58.66 | 33.05 | 19.30 | 17.59 | 47.91 |
| PA3479 | *rhlA* | Rhamnosyltransferase chain A | 18.19 | 84.86 | 6.79 | 20.95 | 13.62 |
| PA3724 | *lasB* | Elastase LasB | 11.07 | 103.15 | 3.31 | 25.49 | 15.78 |
| PA4141 | | Hypothetical protein | 3.80 | 23.59 | 2.69 | 63.25 | 4.67 |
| PA4210 | *phzA1* | Probable phenazine biosynthesis protein | 8.34 | 1788.72 | 6.27 | 13.02 | 63.05 |
| PA4211 | *phzB1* | Probable phenazine biosynthesis protein | 23.71 | 1256.93 | 4.25 | 39.82 | 50.65 |

[a]PAO1 locus tags as determined by sequence homology.
[b]Gene names and functions from *Pseudomonas*.com (86), with *aze* annotation from reference 41.

In addition to the positively regulated genes described above, we sought to determine if any commonalities existed between the genes found to be repressed by RhlR in each isolate. RhlR-repressed regulons varied in size from 1 gene (E125) to 43 genes (E113) and in combination yielded a total of 69 genes across all 5 regulons. However, no genes were shared among all 5 regulons, and only a single gene was found to be shared among any two strains (Fig. S3), consistent with the idea that RhlR is a transcriptional activator (16).

To better understand the scope of RhlR regulation in our cohort of CF isolates, we combined all 5 CF isolate RhlR regulons to create a RhlR panregulon of 227 genes (Table S4). This RhlR panregulon list includes 26 genes not currently annotated in the PAO1 genome; 14 genes are not present in the PAO1 genome, while the remaining 12 are present in the PAO1 genome but not annotated as open reading frames (ORFs). Six of these acquired genes are positively regulated by RhlR in more than one isolate. There are also 3 putative operons controlled by RhlR in this acquired gene list: a 2-gene operon in E125 (287.9771.peg.292 and -293), a separate two-gene operon in E131 (2913690790 and -1), and a 6-gene operon in E113 (2913679565 to -70).

Our observations of variability in the scope of QS regulation among our CF isolate cohort led us to question what genes might represent direct RhlR regulation. We found that 30 of the 201 genes in the RhlR panregulon (28%, genes annotated in PAO1 only) harbor a putative binding site for RhlR upstream of the coding sequence, and an additional 33 genes are in predicted operons with a putative RhlR binding site in the promoter upstream of the first gene (Table S4). Of these 63, 18 directly RhlR-regulated genes are in the RhlR core regulon (90% of the core regulon).

## DISCUSSION

Infection of the CF airways by *P. aeruginosa* presents an intractable clinical problem. A growing body of evidence suggests that adaptation by *P. aeruginosa* to the CF infection niche includes a rewiring of the QS gene regulatory network observed in laboratory strains, with deleterious mutation of *lasR* leaving RhlR as the pivotal QS regulator. Unlike *lasR*, mutation of RhlR in CF isolates is uncommon (22, 34), which implies an evolutionary pressure for maintaining a functional Rhl-QS system as isolates transition to a more chronic phenotype. Thus, an understanding of mechanisms of RhlR liberation

from LasR control, as well as the conserved targets of RhlR regulation, presents an inflection point in efforts to control *P. aeruginosa* infections.

We focused our genetic analysis on a group of LasR-null CF isolates in an attempt to understand how QS might be rewired in lung infections. In our pangenomic analysis, we did not find an obvious link between the common genes lost or gained relative to laboratory strains that could explain this rewiring. Indeed, no known modulator of QS was found to be mutated in all our isolates except *lasR* itself. However, many candidates for this function remain. Mutation of the drug efflux regulator *mexT* was previously shown to facilitate RhlR independence in LasR-null PAO1, although the precise mechanism that leads to increased RhlR activation in that background is still mysterious (37, 38). However, like E90 in our initial description of Las-independent Rhl-QS, the isolates in this study harbor functional *mexT* alleles (33). Other regulators of multidrug efflux were found to be mutated across our isolate cohort (*armR*, *oprR*, and *cmrA*), and it is possible that a combination of mutations other than in *mexT* could yield a mechanistically similar effect.

More broadly, our analysis provides a framework to approach future studies of genome evolution in CF-adapted *P. aeruginosa*. Our study includes isolates from a pediatric study that followed young individuals with CF after acquisition of *P. aeruginosa* infection. Genomic analysis indicated that adaptation is rapid in these populations; all 5 CF isolates in the current study harbored deleterious mutations in 197 genes, many coding for acute virulence factors. In addition to *lasR*, mutation of genes coding for O-antigen biosynthesis, exotoxin A, type III secretion system components, multidrug efflux, and twitching motility was present in our CF isolates compared with the laboratory strain PAO1. Mutation of genes in each of these categories was previously observed during the adaptation of a clonal lineage of infecting *P. aeruginosa* studied in a pediatric patient, followed from 12 months to 96 months of age (22). We also found common mutations in genes coding for S-pyocin production, a bacteriocin mechanism that allows *P. aeruginosa* to toxify neighboring nonkin strains using proteinaceous antimicrobial peptides internalized via siderophore receptors such as FptA (43, 44). Divergence of genes coding for S-pyocins may reflect the pressures of interbacterial competition within infections, as selection may also target other similar competitive mechanisms, such as the phage tail-like R-pyocins or T4SS, in evolving infection isolates (45). The regional isolation of specific lineages of *P. aeruginosa* found in a study of explanted CF lungs provides additional support for this view of isolate pathoadaptation; as bacteria become isolated to a specific long-term niche in the lung, selection may purify mechanisms associated with competitive exclusion (46).

Most descriptions in laboratory strains have found that *lasR* mutation yields a QS-null phenotype (47). There are exceptions: phosphate limitation allows QS regulation in LasR-null laboratory strains through activation of the *pho* signal transduction system (48). However, a variety of LasR-null *P. aeruginosa* isolates (49), including those in this study, engage in QS in a manner that does not require phosphate limitation.

Here, we show that inactivating mutations in *lasR* can be present in multiple isolates that still activate QS target promoters through RhlR and can subsequently produce strong QS phenotypes. Mutation of *lasR* has been previously connected to the early stages of chronic CF infections, and the timing of collection of the EPIC study isolates used here agrees with that notion (22, 50, 51). Studies that focused on trends in *P. aeruginosa* virulence phenotypes associated with worsening lung disease have generally found these phenotypes to be attenuated. Pyocyanin and protease production were associated more with new onset or intermittent infection isolates than chronic isolates in an epidemiological survey that was part of the EPIC study, and a related study associated *lasR* mutation with both these phenotypes and with failure to eradicate *P. aeruginosa* (52, 53). Our isolate cohort exhibited nominal changes in protease activity and significantly increased pyocyanin in every isolate, in contrast to the general trends observed in those earlier studies. Las-independent RhlR control of such phenotypes does not explain this discrepancy, but our data provide an alternate framework

for understanding *lasR* mutation in long-lived infections. What previously was assumed to be an evolutionary path of QS inactivation may instead be a retuning of the QS network in chronic infection.

Our transcriptome analysis exploited these CF *P. aeruginosa* isolates to understand and generalize the scope of RhlR gene regulation. We found 20 genes encoding the production of a minimum of 7 products in common among isolates in our RhlR core regulon analysis. In addition to gene promoters known to be tightly controlled by RhlR (*rhlA*, rhamnolipid biosynthesis), our results confirmed RhlR control of many genes previously described as key virulence factors controlled by Las-dominated QS; *lasB* elastase, hydrogen cyanide biosynthesis (*hcnABC*), and phenazine biosynthesis (*phzA1B1*) are now understood to be regulated at least in part by RhlR. Our results also firmly position the azetidomonamide biosynthesis locus (*aze*, PA3326 to -34) under transcriptional control by RhlR. Indeed, nearly half of the core RhlR regulon is dedicated to these biosynthetic genes. While the primary biosynthetic product of this nonribosomal peptide synthetase (NRPS)-containing cluster has recently been described to be azabicyclene, among other congeners, the biological importance of this novel group of compounds to *P. aeruginosa* is unknown (41, 42). The presence of the gene coding for the $C_4$-HSL signal synthase RhlI in the core regulon provides evidence for a Rhl-QS circuit capable of autoregulatory feedback, an established feature of the LasR-I system (54–56). We are interested in discerning the features of the RhlR-I autoregulatory loop in future investigations of Las-independent QS.

The gene coding for VqsR was the only one coding for a transcriptional regulator discovered in our core RhlR regulon analysis. VqsR has been previously found to be activated directly by LasR (57), and was also shown to provide a general activating effect on the QS circuitry through increased signal production (58). However, a focused study showed that this effect is indirect; VqsR does not directly bind the promoters of any recognized QS circuitry beyond the orphan regulator QscR (59). Although VqsR was also found to directly bind the promoters of a small group of additional genes in that study, that VqsR would repress *qscR* to upregulate the QS circuitry fits with existing evidence surrounding such an antiactivation role for QscR (13, 14, 60). QscR responds to LasI-generated $3OC_{12}$-HSL to directly activate a single linked operon (PA1895 to -1897) and produce a net-repressive effect on QS-controlled genes (14, 60). QscR and its regulatory targets are not mutated in the 5 studied isolates. Our experiments did not evaluate QscR activation dynamics; the role of QscR activation in infections and the mechanism of QS-repressive effects conferred by PA1895 to -1897 remain unclear. Our results expand the control of *vqsR* beyond LasR to also include RhlR and further demonstrate its conservation as a QS target in pathoadapted isolates, but additional studies will be necessary to disentangle the role of this downstream transcriptional activator, as well as QscR, in the interconnected QS networks of *P. aeruginosa*.

We noted several differences in RhlR regulon size and strength of regulation among our tested isolates, suggesting that the conditions of Rhl-QS control are diverse among strains with defective LasR. Strain E113 exhibited the greatest breadth and strength of RhlR control in our transcriptome analysis, with 185 genes activated and an average of 219-fold activation of the core RhlR regulon genes. Pyocyanin production in this strain was 25-fold greater than in PAO1 and essentially absent upon deletion of *rhlR*, consistent with tight regulation of the *phzA1* locus in our transcriptome analysis (>1,000-fold). However, the much smaller RhlR-activated fold change value of approximately 15-fold exhibited in strain E131 was associated with similar pyocyanin phenotypes. This disparity suggests that transcript fold change values may be more indicative of tightness of regulation than reflective of phenotypes.

We were interested in how our RhlR core regulon analysis compared to two previous studies of QS gene regulation in CF isolates. We first compared the core RhlR regulon to that of our previously published analysis of the RhlR regulon of isolate E90 (33). E90 was shown to positively regulate 53 genes in that analysis, and addition of that regulon to our core analysis did not change the 20 core genes regulated by RhlR. We

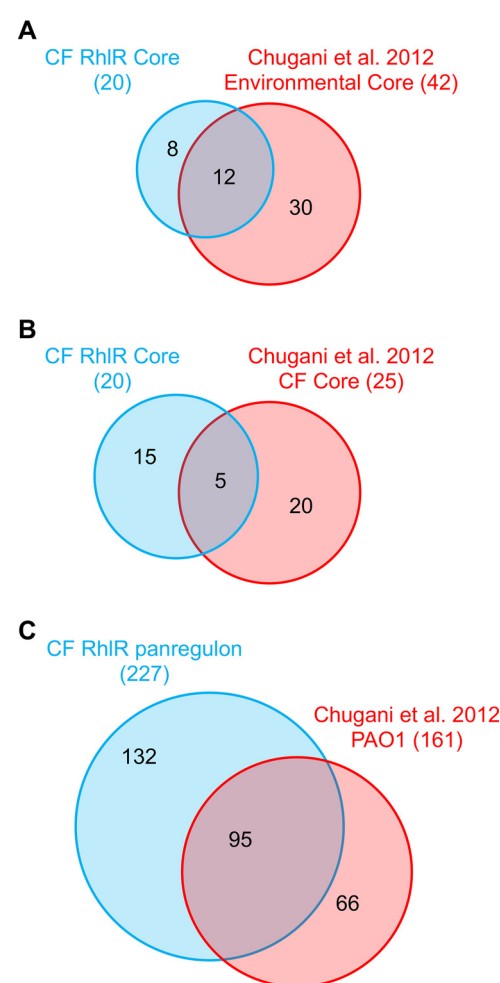

**FIG 5** RhlR regulon comparisons. Comparison of the CF isolate RhlR core regulon determined here with (A) a core QS regulon of 4 environmental strains and PAO1 and (B) a core QS regulon of 2 CF isolates. (C) Comparison of the CF isolate RhlR panregulon with a PAO1 QS regulon. Comparison regulons are from reference 40. Venn diagram lobes in each panel are scaled to approximate proportions.

then compared the RhlR core regulon to the QS regulons reported by Chugani and colleagues for PAO1 and 4 environmental isolates (referred to as the Chugani environmental core; 5 strains, 42 genes) and a core QS regulon among 2 CF isolates (Chugani CF core; 2 strains, 25 genes) (40). The Chugani environmental core shares 12 genes with our RhlR core regulon, including *hcnABC*, *rhlI*, *lasB*, *vqsR*, and much of the *aze* biosynthetic cluster (Fig. 5A). In contrast, the Chugani CF core shares just 5 genes with our RhlR core regulon, including *lasB*, *vqsR*, and 3 genes of the *aze* operon (Fig. 5B).

We combined all 5 CF isolate RhlR regulons to create a RhlR panregulon of 227 genes, providing a more complete scope of RhlR regulation in our isolate cohort (Table S4). Twenty-six of these genes are not present in PAO1, indicating the scope of QS regulatory targets is broader than that identified in laboratory strains. We then compared the remaining 201 genes of the RhlR panregulon to the PAO1 QS transcriptome described in the study by Chugani and colleagues, as that study used comparable conditions in determination of that reference QS regulon (40). In that study, PAO1 was found to positively regulate 161 genes, 95 (59%) of which are represented in our RhlR panregulon (Fig. 5C; Table S4). The remaining 106 RhlR-regulated genes that are annotated in PAO1, but not reported as differentially regulated by QS by Chugani and colleagues, are overrepresented for metabolic resources related to niche adaptation. These include genes involved in anaerobic and microaerophilic growth (*ccoN2*, *ccoO2*,

and *nrdD*), ethanol oxidation (*eraS*), nitrous oxide (*nosY*) and nitrate (*nirJ*) reduction, and components of the oxidative stress response (*fprA* and *soxR*). A principal goal of our study was to discern stable, heritable commonalities in RhlR gene regulation among CF isolates as opposed to condition-specific regulation, which guided our transcriptomics approach of standard laboratory media and conditions. That such niche-specific adaptations are clearly present in the RhlR regulons of our tested isolates under laboratory conditions suggests a robust but adaptable relationship between QS and infections.

We also queried the relationship between the RhlR panregulon genes present in PAO1 and a transcriptomic microarray study that studied multiple sampling time points and multiple configurations of QS-on or -off conditions by Schuster and colleagues (26). Similar to comparison with the Chugani study, we found that 62% of our RhlR-regulated genes (125/201) present in PAO1 were also upregulated in at least one condition in at least one time point in that study. One notable example of the robustness of our approach to defining a panregulon is RhlR regulation of phenazine biosynthesis, where we found that RhlR regulates genes from the *phzA1-G1* (*phz1*) operon and, in 4 of 5 isolates, the *phzA2-G2* (*phz2*) operon. The product of both of these nearly identical (~98% nucleotide identity, 100% residue identity) operons is phenzine-1-carboxylic acid, a precursor to pyocyanin (61). The high sequence identity shared among these two loci has prevented meaningful analysis of differential expression in most previous studies (40, 62). We were able to overcome this challenge through a combination of high read depth and our RNA-seq read-mapping strategy (63, 64). The mechanism of RhlR regulation of *phz2* is unclear. A LasR/RhlR binding site lies upstream of *phz1*, but there is no sequence evidence for direct QS regulation of *phz2*. Analysis of *phz1* and *phz2* expression in both liquid culture and colony biofilms has defined a clear role for *phz2* in controlling phenazine production and coordinate virulence (65), and further investigation will be required to understand how RhlR activates transcription of *phz2*.

Our panregulon also revealed that in multiple isolates, RhlR controls the gene coding for the quinolone monooxygenase PqsH. PqsH has previously been reported to be under LasR but not RhlR control in liquid cultures of laboratory strains (15). PqsH oxidizes the relatively low-affinity PqsR ligand 2-heptyl-4-hydroxyquinoline (HHQ) to produce the relatively high-affinity ligand 2-heptyl-3-hydroxy-4(1H)-quinolone (PQS) (66), providing a potential avenue for RhlR to tune PQS signaling in some CF isolates. A recent transcriptome analysis in colony biofilms suggested that *pqsH* may also be conditionally regulated by RhlR in the laboratory strain PA14 (27). Thus, the environmental history and growth condition of a given isolate might impact how RhlR modulates the PQS system, an area for future studies. Together, our panregulon analyses suggest that while timing of QS regulation may account for some differences in regulon content, adaptation to a specific environment may also have lasting effects on the scope of QS regulation.

We conclude that clear commonalities exist among Las-independent Rhl-QS systems in CF-adapted *P. aeruginosa*. Twenty genes encoding 7 products, including potent virulence factors, were regulated by RhlR in our analysis of 5 CF isolates exhibiting this circuitry. Our results position RhlR as an independent QS regulator of virulence genes in *P. aeruginosa*, highlighting the potential for this QS receptor as a therapeutic target (67).

## MATERIALS AND METHODS

**Isolate selection, growth, and characterization.** Bacterial strains and plasmids used in this study are listed in Table S5. The CF infection isolates were originally collected from oropharyngeal and sputum samples from patients 5 to 12 years of age as part of the Early *Pseudomonas* Infection Control (EPIC) observational study (28, 52). Beginning with a cohort of 34 LasR-null isolates that produce $C_4$-HSL, we narrowed our analysis to those that both accepted reporter plasmids (26 isolates) and also allowed engineering of *rhlR* (9 isolates); we selected 5 of these isolates for further testing. Routine cultures were maintained on Luria-Bertani (LB) agar medium at 37°C or in LB broth buffered with 50 mM 3-(*N*-morpholino)propanesulfonic acid (MOPS) at pH 6.8 (LB+MOPS) in 18-mm borosilicate glass culture tubes with 250 rpm orbital shaking at 37°C, unless otherwise noted. When appropriate, gentamicin was added at 10 μg/mL for *Escherichia coli* (Gm10) or 100 μg/mL for *P. aeruginosa* (Gm100). For synthetic signal

addition experiments, 2 $\mu$M *N*-3-oxo-dodecanoyl-L-homoserine lactone (3OC$_{12}$-HSL) or 10 $\mu$M *N*-butanoyl-L-homoserine lactone (C$_4$-HSL) (Cayman Chemical) was added.

**Transcriptional fusions and phenotypic analysis.** LasR and RhlR transcriptional activity was determined using P$_{lasI}$-*gfp* or P$_{rhlA}$-*gfp* transcriptional fusion reporter plasmids, respectively, as described previously (24). Plasmids were introduced into electrocompetent *P. aeruginosa* cells prepared using repeated 300 mM sucrose washes, selected on LB Gm100 agar, and verified by PCR (68). For endpoint experiments, individual gentamicin-resistant colonies were inoculated into LB+MOPS broth with or without synthetic signal, grown at 37°C with 250 rpm shaking, and then harvested at 18 h. Triplicate aliquots (200 $\mu$L each) were then transferred to black-walled chimney-welled fluorescence microtiter plates with optically clear bottoms (Greiner Bio-one) and assayed in a BioTek Synergy HI microplate reader for optical density at 600 nm (OD$_{600}$) and GFP-derived fluorescence ($\lambda_{excitation}$ = 485 nm, $\lambda_{emission}$ = 535 nm). For kinetic experiments, from overnight cultures of isolated colonies in LB+MOPS broth, cells were subcultured 1:100 into fresh LB+MOPS Gm100, grown to mid logarithmic phase (OD$_{600}$ of 0.2 to 0.8, 1-cm path length), and then inoculated into 200 $\mu$L LB+MOPS broth at a starting OD$_{600}$ of 0.01 in triplicate in fluorescence microtiter plates. Plates were then incubated with orbital shaking at 37°C in a plate reader with OD$_{600}$ and GFP measurements every 15 min for 16 h. Reported fluorescence was normalized to OD$_{600}$, and fluorescence derived from each strain with a promoterless control reporter plasmid of the same backbone was subtracted to account for background fluorescence. All experiments were performed with a minimum of biological triplicates.

Pyocyanin production was determined after 18 h of growth with shaking at 37°C in pyocyanin production medium (PPM) as previously described (24). Extracellular protease production was determined by patching isolated colonies onto skim milk agar plates (1/4-strength LB broth–4% [wt/vol] skim milk–1.5% [wt/vol] agar; 20 mL in 150-mm petri dishes) (31), followed by incubation at 37°C for 48 h, and then the zone of proteolytic clearing was measured using a ruler from the edge of the colony. A minimum of 3 biological replicates were performed, and laboratory strain controls were included on every plate.

***rhlR* mutant construction.** Markerless in-frame *rhlR* deletion mutants were made using homologous recombination-mediated allelic exchange as previously described (69, 70). Our approach improves allelic exchange accuracy and efficiency in clinical isolates through use of deletion plasmids that harbor approximately 800 bp of native strain-specific sequence as homologous ends both upstream and downstream of the target locus *rhlR*. Strain-specific *rhlR* deletion plasmids were constructed using *E. coli*-mediated assembly of PCR-amplified products (71). Initial native *rhlR* sequences were determined using PCR amplification and targeted Sanger sequencing (Genewiz; Azenta) of 4 overlapping fragments covering an ~2.5-kb region encompassing the *rhlR* locus, which were then used to design sequence-specific deletion allele construction primers. Allele fragments were generated using the high-fidelity PrimeSTAR Max polymerase (TaKaRa Bio), assembled into strain-specific pEXG2 plasmids in *E. coli* DH5$\alpha$ (New England Biolabs), and introduced into each isolate using *sacB*-mediated sucrose counterselection as previously described (70, 71). Deletion was confirmed using PCR amplification and targeted sequencing.

**Genomic analysis.** High-molecular-weight (HMW) genomic DNA was isolated from liquid cultures using the Qiagen Genomic-tip 20/G kit (Qiagen). DNA was then subjected to a hybrid approach of both short- and long-read DNA sequencing to achieve high-confidence complete *de novo* genome assemblies, as we have reported previously (33). Short reads (Illumina MiSeq, TruSeq v3, 300 bp paired end; Illumina) were groomed using FastQC (Babraham Bioinformatics; Babraham Institute) in combination with Trimmomatic (v0.36; adapter trimming, paired reads only; Phred score cutoff = 15) (72). After base-calling and demultiplexing using Guppy (v3.1.5; Oxford Nanopore), long reads (EXP-PBC001, EXP-NBD114 libraries, Nanopore MinION, R9.4.1 pores; Oxford Nanopore) were groomed using NanoPack (NanoPlot v1.27.0; NanoQC v0.9.1) and Porechop (v0.2.4) (73, 74). The hybrid assembly mode in the Unicycler pipeline (including SPAdes v3.13.0, Racon v1.4.3, and Pilon) was used to assemble and close genomes prior to annotation using the RAST pipeline (75–78). General genomic features are summarized in Table S1.

Comparative genomic analysis was achieved using a combination of established tools. Pangenomic analysis was conducted using complete genomes in the program "anvi-pan-genome" in the anvi'o software ecosystem (79). PAO1 genes not present in all isolates (74) were discovered using the "anvi-display-pan" program. Variant analysis was conducted using Illumina short reads compared against reference strains (PAO1, accession no. NC_002516; PA14, accession no. GCF_000014625.1) using the StrandNGS SNP, CNV, and SV pipelines (v3.3.1; Strand Life Sciences). Variants were defined with a variant read frequency of >90%, and indels were defined as <100 bp. We defined variants as present in cohort isolates but not laboratory strains by first comparing each genome to PAO1, followed by Venn analysis to determine a core group of variants (1,682), followed by filtering against PA14 variants to yield a final list of 123 genes.

We conducted a phylogenetic analysis using the comparative genomics suite available through the Joint Genome Institute Integrated Microbial Genomes and Microbiomes—Expert Review (JGI IMG/MER) web tool analysis suite. The phylogeny was constructed using hierarchical clustering in the Taxonomy-Genus mode, and the presence/absence of GC-1 and -2 in genomes was determined using the genomic Conserved Neighborhood analysis tool.

**Transcriptomic analysis.** We determined the extent of QS-controlled gene regulation in our cohort of 5 CF isolates that exhibit Las-independent Rhl-QS activity using an RNA-seq-based transcriptome analysis. We compared engineered *rhlR* deletion strains (QS-off condition) to isogenic parent strains supplemented with both synthetic C$_4$-HSL (10 $\mu$M) and 3OC$_{12}$-HSL (2 $\mu$M) to ensure QS activation (QS-on). We did so because, while isolates in our cohort do not respond to 3OC$_{12}$-HSL through LasR (Fig. 2), the

orphan LuxR homolog QscR is still intact and likely binds to exogenous $3OC_{12}$-HSL from neighboring bacteria in the context of infections. QscR has been shown to bind and activate a single linked operon beginning with PA1897 (14), and its role in Rhl-QS regulation in these isolates is unknown.

Beginning with isolated colonies, we grew each strain overnight in 3 mL LB+MOPS in 18-mm culture tubes at 37°C with 250-rpm shaking. Saturated cultures were then diluted to an $OD_{600}$ of 0.001 in 25 mL LB+MOPS medium (with AHL signal addition in QS-on samples) in 250-mL aerobic baffled flasks and incubated at 37°C with 250 rpm shaking. Total RNA was harvested from approximately $1 \times 10^9$ cells in the transition from logarithmic to stationary phase ($OD_{600}$, 2.0). Total RNA was preserved and purified using RNAprotect bacterial reagent and an RNeasy minikit with Qiazol (Qiagen) as previously described (40). RNA was quantified using Qubit fluorometric quantitation (Thermo Fisher Scientific), and the lack of RNA degradation was confirmed using NorthernMax-Gly glyoxal gel electrophoresis (Thermo Fisher Scientific) and an Agilent 2100 Bioanalyzer (Agilent). rRNA depletion, library generation, and >20 million 150-bp paired-end Illumina HiSeq reads were generated for each sample commercially (Genewiz; Azenta). Reads were groomed using FastQC (Babraham Bioinformatics, Babraham Institute) and Trim Galore! (v0.4.3; https://github.com/FelixKrueger/TrimGalore). We mapped reads from 3 biological replicates to the strain-specific mapping references generated earlier using the Subread/featureCounts suite of command line tools, and conducted differential expression (DE) analyses in the R statistical environment using strain-specific data sets in the DESeq2 statistical package with a $\log_2$ fold change cutoff of 1 and a false discovery rate (FDR) of 0.05 (63, 64, 80). We used an all-by-all BLASTp comparison to determine the common gene names associated with targets in PAO1, and where possible, we used those names here (81). Genes with BLASTp top hit bit scores less than 50 were considered nonhomologous to any PAO1 gene. *rhlR* was discarded from analysis after computation of DE, as this is a result of our engineered knockouts in the QS-off condition. *qscR* and the single linked target of QscR, the operon spanning PA1897 to -91, were also discarded from analysis, as differential regulation of these genes is due to synthetic $3OC_{12}$-HSL addition in the QS-on condition, as discussed above. Venn analysis and core figure generation were conducted using the online tool provided by Bioinformatics & Evolutionary Genomics, University of Ghent, Belgium (https://bioinformatics .psb.ugent.be/webtools/Venn/).

**Promoter analysis.** A global position analysis has not yet been published for RhlR, but a consensus promoter sequence for binding of LuxR homologs has been determined to have dyad symmetry of $N_2CTN_{12}AGN_2$ centered 38 to 45 bp upstream of the transcriptional start site (TSS) (82, 83). LasR binding sites have been explored in a global position analysis using chromatin immunoprecipitation combined with DNA microarrays (ChIP-chip) and through direct *in vitro* binding in electrophoretic mobility shift assays (84). LasR and RhlR share LuxR homology and have been shown to regulate many of the same promoters, so we assumed that a *las* or *lux* box recognition site may also be directly regulated by RhlR ("*las/rhl*-box"). We cross-referenced our RhlR panregulon with a previous study that mapped the transcription start sites of *P. aeruginosa* strain PA14 to determine whether a gene is predicted to lie in a solely transcribed ORF or if the gene is the first in an operon (85). We then searched the PRODORIC, CollecTF, and PseudoCAP databases to determine if a RhlR-regulated gene, or the first gene in an operon containing a RhlR-regulated gene, has a putative *las/rhl* box (86).

**Data availability.** CF isolate genomes are publicly available through the Integrated Microbial Genomes/Expert Review database (Joint Genome Institute) under the following genome IDs: E104, 2913679597; E113, 2913673089; E125, 2870682680; E131, 2913685979; E167, 2870695665. Genome sequences and transcriptomic data (including raw RNA-seq data) associated with this study are also publicly available through NCBI and the NCBI Gene Expression Omnibus (GEO) under the following accession numbers: E104, PRJNA784987 and GSE198534; E113, PRJNA814681 and GSE198716; E125, PRJNA814691 and GSE198717; E131, PRJNA814694 and GSE198718; E167, PRJNA814703 and GSE198719.

## SUPPLEMENTAL MATERIAL

Supplemental material is available online only.

**FIG S1**, PDF file, 0.04 MB.

**FIG S2**, PDF file, 0.8 MB.

**FIG S3**, PDF file, 0.5 MB.

**TABLE S1**, PDF file, 0.02 MB.

**TABLE S2**, PDF file, 0.05 MB.

**TABLE S3**, PDF file, 0.1 MB.

**TABLE S4**, PDF file, 0.2 MB.

**TABLE S5**, PDF file, 0.05 MB.

## ACKNOWLEDGMENTS

This work was supported by U.S. Public Health Service (USPHS) grant R01GM125714 (to A.A.D.). A.A.D. was also supported by the Burroughs-Wellcome Fund (grant 1012253) and Doris Duke Charitable Foundation (grant 2017073). K.L.A. was supported by a postdoctoral fellowship from the Cystic Fibrosis Foundation (ASFAHL19F0). We

acknowledge core support from USPHS grant P30DK089507 and the Cystic Fibrosis Foundation (grants SINGH15R0 and R565 CR11).

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
