## [Reviewer comments · mSystems]

Genetic and transcriptomic characteristics of RhIR-dependent quorum sensing in cystic fibrosis isolates of *Pseudomonas aeruginosa*

Kyle Asfahl, Nicole Smalley, Alexandria Chang, and Ajai Dandekar

Corresponding Author(s): Ajai Dandekar, University of Washington

Review Timeline:

Submission Date:	February 3, 2022
Editorial Decision:	February 22, 2022
Revision Received:	March 17, 2022
Accepted:	March 23, 2022

Editor: Mark Mandel

Reviewer(s): Disclosure of reviewer identity is with reference to reviewer comments included in decision letter(s). The following individuals involved in review of your submission have agreed to reveal their identity: Joshua D ShROUT (Reviewer #2)

Transaction Report:

DOI: <https://doi.org/10.1128/msystems.00113-22>

February 22, 2022

Dr. Ajai A Dandekar
University of Washington
Medicine
1959 NE Pacific Street
Box 356522
Seattle, Washington 98195

Re: mSystems00113-22 (Genetic and transcriptomic characteristics of RhlR-dependent quorum sensing in cystic fibrosis isolates of *Pseudomonas aeruginosa*)

Dear Dr. Ajai A Dandekar:

Thank you for submitting your manuscript to mSystems. We have completed our review and I am pleased to inform you that, in principle, we expect to accept it for publication in mSystems. However, acceptance will not be final until you have adequately addressed the reviewer comments. Both reviewers were impressed with the quality of both the scientific work and the writing. They both offered comments to improve aspects of the manuscript, including suggestions to improve clarity and focus (especially given the large number of genes described). I do not expect that addressing any of these comments will require acquisition of new data, though a couple of them may require additional analysis on data that you have. Consistent with ASM Journals' Data Availability guidelines, please ensure that the NCBI accession is made public prior to submission of the modified manuscript.

Preparing Revision Guidelines

- Point-by-point responses to the issues raised by the reviewers in a file named "Response to Reviewers," NOT IN YOUR COVER LETTER.
- Upload a compare copy of the manuscript (without figures) as a "Marked-Up Manuscript" file. This should clearly show all changes made since the previous submission.
- Each figure must be uploaded as a separate file, and any multipanel figures must be assembled into one file.
- Manuscript: A .DOC version of the revised manuscript
- Figures: Editable, high-resolution, individual figure files are required at revision, TIFF or EPS files are preferred

Sincerely,

Mark Mandel

Editor, mSystems

Journals Department
Reviewer comments:

Reviewer #1 (Comments for the Author):

Asfahl et al. use transcriptomic analyses of 5 cystic fibrosis clinical isolates of *Pseudomonas aeruginosa* to determine the core RhIR-dependent regulon in the absence of its own transcriptional activator, LasR. Previous investigations, both on laboratory strains and clinical isolates under different growth conditions, established the importance of RhIR in quorum sensing progression and long-term chronic infections. This paper elegantly builds upon these studies. The important insight from this paper is determining which *P. aeruginosa* genes are regulated by RhIR across different isolates to establish why certain genes are upregulated and how these expression changes might be selected for in the host lung. Indeed, the manuscript relies on a single large-scale transcriptomics approach with phenotypic analyses to draw broad conclusions about signaling in the host. The experiments were well-controlled, and their conclusions were consistent with their results.

The data presented here are important to the field of quorum sensing and are of general interest to the wider field of bacteriology because of its insights into the regulation of signal transduction and pathogenesis in an important human pathogen. The findings are also of great interest to clinical microbiologists studying infections of patients suffering with cystic fibrosis and other pulmonary disorders.

I have a few issues that I would like to see addressed before publication. These are related to contextualizing the results in the light of other recent findings about RhIR as well as clarifying some aspects of their methodology and interpretation of the results.

1. The authors do a good job of introducing *Pseudomonas* QS. It's not an easy thing to do for an introduction. However, additional information should be included after line 57:
 - a. In addition to be pqsABCDE being activated by LasR, it is repressed by RhIR. PMID: 25225275
 - b. PQS, specifically, pqsE regulates RhIR activity as a transcription factor. This should be an important factor to note when discussing RhIR function. PMID: 35019777; PMID: 30224496; PMID: 31194839; PMID: 33793200; PMID: 32457239; PMID: 18776012
 - i. It would be beneficial to alter Figure 1 to reflect this more accurate portrayal of quorum sensing, especially considering that the authors have decided to include qscR in the schematic.
 - c. I believe it is also worth mentioning that lasR null laboratory strains can also be bypassed under different growth conditions: PMID: 33288622. It would be appropriate to include this in the paragraph starting with line 318.
2. On line 111-112, isolates were said to be selected based on if they were amenable to genetic manipulation. In the Methods section, it might be beneficial to discuss how this was performed and the steps that were taken that might be different from traditional cloning in PAO1 or PA14.
3. Related to Table 1: the authors should discuss how or why these particular amino acid substitutions are detrimental to LasR function in these isolates.
4. On line 192-193, the authors state, "no other gene with an established direct regulatory link to the Las, RhI, or PQS QS systems was found to be mutated." Can the authors elaborate on how those criteria were established?
5. Related to line 193-196: an article I cite above (PMID: 33288622), the pho signal transduction system can regulate RhIR expression. Can they elaborate on whether this signaling cascade was altered?
6. I have issues with lines 205-221. This criticism is not based on the science, but the way the information is displayed and discussed, which I think is a shortcoming of the manuscript. There are a lot of data here and a lot to interpret, which results in only superficial analysis. I would consider cutting most of this section.
7. Line 228: For clarity, please specify the ligand that was added.
8. Related to Table 2: why aren't the entirety of both phz operons represented in the core regulon? The expression of phzC1-E1 should not be that markedly different from phzA1 or phzB1 especially when you consider there's ~2000-fold change occurring in certain isolates.
9. The Parsek lab has shown that psl and pel are highly expressed in clinical samples taken from a CF lung. Can the authors speak to why they do not observe increases in pel or psl expression as part of their core regulon?
10. Line 408, when discussing PqsH: the link between RhIR and PQS signaling has already been established and should be cited using the articles listed in 1b.

Reviewer #2 (Comments for the Author):

The study reports upon Rhl quorum sensing system characteristics of the bacterium *Pseudomonas aeruginosa*. While the "textbook" and canonical understanding of *P. aeruginosa* quorum sensing relies upon signaling of the Las system to cue the Rhl system (and more), several prior studies have shown that cystic fibrosis lung isolates of *P. aeruginosa* are non-functional for the Las system. (This point is well-noted by the authors.) Thus, this manuscript addresses a key area to understand this notable collective group of Las mutants-which activities are still QS regulated? This is important work from both ecological and clinical points-of-view.

This manuscript contains substantial information about Rhl quorum sensing in strains that are LasR deficient. While several of the specific findings presented will be appreciated best by those with strong knowledge of *P. aeruginosa* quorum sensing-the manuscript does an excellent job of contextualizing these results. While the authors correctly self-identify that further work is needed to elucidate mechanism and build a deeper understanding of Rhl quorum sensing in LasR-mutants, there is clearly a basis upon which further study can be conducted down multiple specific paths (established by this work).

The authors are to be commended on preparing such a well-written manuscript. Such a treat to review!

Specific points:

Line 108. Can some accounting of how many isolates were discounted, as a result of the criteria stated, be given? It is mostly just curiosity, but the characterization is useful, to delineate how one gets from 31 isolates to the 5 used here.

Line 113. Please give some parameters for the measured C4-HSL concentrations. These are 24h planktonic cultures? Or what? Also, is the level of C12-HSL production also known for these strains?

Line 142. The statement about LasB in PAO1 is correct. Thus, wouldn't a more useful control on Fig 2D be to show PAO1 lasR and PAO1 rhlR single mutation effects on elastase activity?

Line 222. This section is really great.

Line 351. In this paragraph, or separately, the authors should point out that QscR is not really probed in this study-thus, this is also a future need of research? It is my recollection that prior work showed some differential expression of QscR regulated genes in response to C4-HSL and C12-oxo-HSL. Thus, is there also rewiring of QscR actions in these LasR mutants? And what of PQS activation?

Asfahl *et al.* use transcriptomic analyses of 5 cystic fibrosis clinical isolates of *Pseudomonas aeruginosa* to determine the core RhIR-dependent regulon in the absence of its own transcriptional activator, LasR. Previous investigations, both on laboratory strains and clinical isolates under different growth conditions, established the importance of RhIR in quorum sensing progression and long-term chronic infections. This paper elegantly builds upon these studies. The important insight from this paper is determining which *P. aeruginosa* genes are regulated by RhIR across different isolates to establish why certain genes are upregulated and how these expression changes might be selected for in the host lung. Indeed, the manuscript relies on a single large-scale transcriptomics approach with phenotypic analyses to draw broad conclusions about signaling in the host. The experiments were well-controlled, and their conclusions were consistent with their results.

The data presented here are important to the field of quorum sensing and are of general interest to the wider field of bacteriology because of its insights into the regulation of signal transduction and pathogenesis in an important human pathogen. The findings are also of great interest to clinical microbiologists studying infections of patients suffering with cystic fibrosis and other pulmonary disorders.

I have a few issues that I would like to see addressed before publication. These are related to contextualizing the results in the light of other recent findings about RhIR as well as clarifying some aspects of their methodology and interpretation of the results.

1. The authors do a good job of introducing *Pseudomonas* QS. It's not an easy thing to do for an introduction. However, additional information should be included after line 57:
 - a. In addition to be *pqsABCDE* being activated by LasR, it is repressed by RhIR. PMID: 25225275
 - b. PQS, specifically, *pqsE* regulates RhIR activity as a transcription factor. This should be an important factor to note when discussing RhIR function. PMID: 35019777; PMID: 30224496; PMID: 31194839; PMID: 33793200; PMID: 32457239; PMID: 18776012
 - i. It would be beneficial to alter Figure 1 to reflect this more accurate portrayal of quorum sensing, especially considering that the authors have decided to include *qscR* in the schematic.
 - c. I believe it is also worth mentioning that *lasR* null laboratory strains can also be bypassed under different growth conditions: PMID: 33288622. It would be appropriate to include this in the paragraph starting with line 318.
2. On line 111-112, isolates were said to be selected based on if they were amenable to genetic manipulation. In the Methods section, it might be beneficial to discuss how this was performed and the steps that were taken that might be different from traditional cloning in PAO1 or PA14.
3. Related to Table 1: the authors should discuss how or why these particular amino acid substitutions are detrimental to LasR function in these isolates.

4. On line 192-193, the authors state, “no other gene with an established direct regulatory link to the Las, Rhl, or PQS QS systems was found to be mutated.” Can the authors elaborate on how those criteria were established?
5. Related to line 193-196: an article I cite above (PMID: 33288622), the *pho* signal transduction system can regulate RhIR expression. Can they elaborate on whether this signaling cascade was altered?
6. I have issues with lines 205-221. This criticism is not based on the science, but the way the information is displayed and discussed, which I think is a shortcoming of the manuscript. There are a lot of data here and a lot to interpret, which results in only superficial analysis. I would consider cutting most of this section.
7. Line 228: For clarity, please specify the ligand that was added.
8. Related to Table 2: why aren't the entirety of both *phz* operons represented in the core regulon? The expression of *phzC1-E1* should not be that markedly different from *phzA1* or *phzB1* especially when you consider there's ~2000-fold change occurring in certain isolates.
9. The Parsek lab has shown that *psl* and *pel* are highly expressed in clinical samples taken from a CF lung. Can the authors speak to why they do not observe increases in *pel* or *psl* expression as part of their core regulon?
10. Line 408, when discussing PqsH: the link between RhIR and PQS signaling has already been established and should be cited using the articles listed in 1b.

Response to Reviewers

Reviewer #1 (Comments for the Author):

1. The authors do a good job of introducing Pseudomonas QS. It's not an easy thing to do for an introduction. However, additional information should be included after line 57:

a. In addition to pqsABCDE being activated by LasR, it is repressed by RhIR. PMID: 25225275

Response: We agree and have added reference to the Brouwer study on line 63.

b. PQS, specifically, pqsE regulates RhIR activity as a transcription factor. This should be an important factor to note when discussing RhIR function. PMID: 35019777; PMID: 30224496; PMID: 31194839; PMID: 33793200; PMID: 32457239; PMID: 18776012

Response: As with comment 1a, we have revised this passage to include more detail regarding the interconnected nature of P. aeruginosa QS, particularly with regard to RhIR and the PQS system. We have included the suggested references (now lines 58-63).

i. It would be beneficial to alter Figure 1 to reflect this more accurate portrayal of quorum sensing, especially considering that the authors have decided to include qscR in the schematic.

Response: We have modified Figure 1 to reveal extra detail regarding the regulatory interactions between RhIR and the PQS system. We have also added an additional phrase on line 55 to better connect the text regarding QscR to Figure 1.

c. I believe it is also worth mentioning that lasR null laboratory strains can also be bypassed under different growth conditions: PMID: 33288622. It would be appropriate to include this in the paragraph starting with line 318.

Response: We appreciate the reviewer pointing out this reference – we have added discussion of that paper in the specified section (now lines 291-295).

2. On line 111-112, isolates were said to be selected based on if they were amenable to genetic manipulation. In the Methods section, it might be beneficial to discuss how this was performed and the steps that were taken that might be different from traditional cloning in PAO1 or PA14.

Response: We added additional discussion in the Methods section (rhIR mutant construction, beginning line 461) of our modifications to the allelic exchange technique that improve efficiency in clinical isolates.

3. Related to Table 1: the authors should discuss how or why these particular amino acid substitutions are detrimental to LasR function in these isolates.

Response: We have added this detail to the main text (lines 110-112).

4. On line 192-193, the authors state, "no other gene with an established direct regulatory link to the Las, RhI, or PQS QS systems was found to be mutated." Can the authors elaborate on how those criteria were established?

Response: We added detail in the form of the list of loci evaluated, which includes the QS biosynthetic machinery and direct anti-activators of QS (now appears on line 177-179).

5. Related to line 193-196: an article I cite above (PMID: 33288622), the pho signal transduction system can regulate RhIR expression. Can they elaborate on whether this signaling cascade was altered?

Response: Our study did not specifically test whether the pho signal transduction system was altered in the tested clinical isolates. Neither *phoB* or *phoR* were among the genes mutated across all tested isolates (Table S2). Regardless, we acknowledge the role of the *pho* pathway in activating the Rhl system under certain conditions and have added discussion of this feature and this reference to the discussion pursuant to comment 1c above.

6. I have issues with lines 205-221. This criticism is not based on the science, but the way the information is displayed and discussed, which I think is a shortcoming of the manuscript. There are a lot of data here and a lot to interpret, which results in only superficial analysis. I would consider cutting most of this section.

Response: We have shorted this section considerably for brevity (we removed the passage that was lines 216-221 in the original manuscript).

7. Line 228: For clarity, please specify the ligand that was added.

Response: We have specified the added ligands on what is now line 208. The rationale for adding both ligands remains in the Methods section.

8. Related to Table 2: why aren't the entirety of both phz operons represented in the core regulon? The expression of *phzC1-E1* should not be that markedly different from *phzA1* or *phzB1* especially when you consider there's ~2000-fold change occurring in certain isolates.

Response: The reviewer raises an interesting point regarding the phenazine loci, and differences in expression among the genes within the *phzA1B1C1D1E1F1G1* operon have multiple sources. First, the two phenazine-1-carboxylic acid biosynthesis operons (*phzA1*, PA4210; *phzA2*, PA1899) show very high identity (>98% nucleotide), a feature that has thwarted many previous analyses as it has been difficult to map reads to these loci unambiguously. Our approach (both the de novo sequencing/mapping and the chosen software) allow differentiation between the two loci, but ambiguous reads are still discarded from analysis which can affect overall read totals for individual higher-identity loci (particularly *phzE1/E2* and *phzF1/F2*). Second, representation in the core regulon requires genes have both statistically significant differential expression and sufficient fold-change in all isolates tested. In the case of *phzC1-E1* or even the entire *phzA2* locus, these genes were indeed significantly controlled by RhlR in several isolates, even with very high fold-change values, but the fold-change simply did not cross the threshold of significance for all five isolates. This phenomenon is apparent in the RhlR panregulon (Table S4) and we have added to the discussion (lines 389-401) to more clearly explain these features of our analysis.

9. The Parsek lab has shown that *psl* and *pel* are highly expressed in clinical samples taken from a CF lung. Can the authors speak to why they do not observe increases in *pel* or *psl* expression as part of their core regulon?

Response: The reviewer is correct that both *psl* and *pel* are frequently found to be highly expressed in clinical isolates, and both loci appear highly expressed in our isolates (data not shown). However, we are not aware of any data that indicate their transcriptional control by RhlR, including in our study, a feature which would be required for them to be present in our regulatory analysis.

10. Line 408, when discussing PqsH: the link between RhlR and PQS signaling has already been established and should be cited using the articles listed in 1b.

Response: We appreciate the reviewer's suggestion to include the citations in comment 1b, which we did include in the introductory passage as requested. The passage

regarding PqsH on line 408 in the original manuscript specifically addresses direct RhIR regulation of pqsH, a feature that has only previously been described in the literature once, and under different (colony biofilm) conditions (PMID: 31194839). We have edited the passage on line 408 (lines 407-410 in the revised manuscript) to include this feature and the quoted reference.

Reviewer #2 (Comments for the Author):

The authors are to be commended on preparing such a well-written manuscript. Such a treat to review!

Thank you!

Specific points:

Line 108. Can some accounting of how many isolates were discounted, as a result of the criteria stated, be given? It is mostly just curiosity, but the characterization is useful, to delineate how one gets from 31 isolates to the 5 used here.

Response: We have added a passage to the Methods section that better describes in detail how many isolates passed each of the criteria (lines 424-426 in the revised manuscript).

Line 113. Please give some parameters for the measured C4-HSL concentrations. These are 24h planktonic cultures? Or what? Also, is the level of C12-HSL production also known for these strains?

Response: We have added detail to Table 1 to include both the parameters of the experiments, as well as the 3OC12-HSL concentrations produced by these strains.

Line 142. The statement about LasB in PAO1 is correct. Thus, wouldn't a more useful control on Fig 2D be to show PAO1 lasR and PAO1 rhIR single mutation effects on elastase activity?

Response: We appreciate the reviewer's attention to this detail. We have added data for the PAO1 lasR and rhIR single mutants to Figure 2D, along with exposition of the results in lines 143-145 of the revised manuscript.

Line 222. This section is really great.

Response: Thank you!

Line 351. In this paragraph, or separately, the authors should point out that QscR is not really probed in this study-thus, this is also a future need of research? It is my recollection that prior work showed some differential expression of QscR regulated genes in response to C4-HSL and C12-oxo-HSL. Thus, is there also rewiring of QscR actions in these LasR mutants? And what of PQS activation?

Response: Point taken. We have added discussion of QscR to the revised manuscript (paragraph beginning line 328). We are not aware of a previous study that evaluated differential expression of QscR-regulated genes in response to C4- and 3OC12-HSL (there was no difference attributable to signal in the Schuster et al 2003 study).

PQS activation is now discussed as an area for future research in lines 409-410 of the revised manuscript.

March 23, 2022

Dr. Ajai A Dandekar
University of Washington
Medicine
1959 NE Pacific Street
Box 356522
Seattle, Washington 98195

Re: mSystems00113-22R1 (Genetic and transcriptomic characteristics of RhlR-dependent quorum sensing in cystic fibrosis isolates of *Pseudomonas aeruginosa*)

Dear Dr. Ajai A Dandekar:

Your manuscript has been accepted, and I am forwarding it to the ASM Journals Department for publication. For your reference, ASM Journals' address is given below. Before it can be scheduled for publication, your manuscript will be checked by the mSystems production staff to make sure that all elements meet the technical requirements for publication. They will contact you if anything needs to be revised before copyediting and production can begin. Otherwise, you will be notified when your proofs are ready to be viewed.

Publication Fees:

We recognize that the video files can become quite large, and so to avoid quality loss ASM suggests sending the video file via <https://www.wetransfer.com/>. When you have a final version of the video and the still ready to share, please send it to mSystems staff at mSystems@asmusa.org.

For mSystems research articles, if you would like to submit an image for consideration as the Featured Image for an issue, please contact mSystems staff at mSystems@asmusa.org.

Sincerely,

Mark Mandel
Editor, mSystems

Journals Department
Table S4: Accept
Table S2: Accept
Table S5: Accept
Table S3: Accept
Figure S3: Accept
Figure S1: Accept
Figure S2: Accept
Table S1: Accept